nanotechnology/materials science

MXene, exfoliation, fluid flow

**Author for correspondence:**
Colin L. Raston
e-mail: colin.raston@flinders.edu.au

This article has been edited by the Royal Society of Chemistry, including the commissioning, peer review process and editorial aspects up to the point of acceptance.

# Continuous flow vortex fluidic-mediated exfoliation and fragmentation of two-dimensional MXene

Ahmed Hussein Mohammed Al-antaki[1,3], Suela Kellici[4], Nicholas P. Power[5], Warren D. Lawrance[2] and Colin L. Raston[1]

[1]Flinders Institute for Nanoscale Science and Technology, College of Science and Engineering, and [2]College of Science and Engineering, Flinders University, Adelaide, South Australia 5042, Australia
[3]Department of Chemistry, Faculty of Sciences, University of Kufa, Kufa, Najaf, Iraq
[4]School of Engineering, London South Bank University, 103 Borough Road, London SE1 0AA, UK
[5]School of Life, Health and Chemical Sciences, The Open University, Walton Hall, Milton Keynes MK7 6AA, UK

AHMA, 0000-0001-8576-3805; SK, 0000-0003-1387-2345; NPP, 0000-0002-4630-7580; CLR, 0000-0003-4753-0079

MXene ($Ti_2CT_x$) is exfoliated in a vortex fluidic device (VFD), as a thin film microfluidic platform, under continuous flow conditions, down to *ca* 3 nm thin multi-layered two-dimensional (2D) material, as determined using AFM. The optimized process, under an inert atmosphere of nitrogen to avoid oxidation of the material, was established by systematically exploring the operating parameters of the VFD, along with the concentration of the dispersed starting material and the choice of solvent, which was a 1 : 1 mixture of isopropyl alcohol and water. There is also some fragmentation of the 2D material into nanoparticles *ca* 68 nm in diameter.

## 1. Introduction

MXenes are a unique class of two-dimensional (2D) material, first reported in 2011 [1]. They are transition metal carbides and carbonitrides and have a number of potential applications, including in biology [2–5], batteries [6–8], electronic devices [9,10] and supercapacitors [10,11]. They are prepared from $M_{n+1}AX_n$ phases via etching the A layers from the laminar material, where M is a transition metal (Ti, Zr, Nb, V, Ta or Mo), A is a main group element, mostly Al or Si, and X is C and/or N, and *n* is 1, 2 or 3. Substituting Al or Si with functional groups, T (–O, –OH, –F), either side of the sheets ($M_{n+1}X_nT_x$) imparts greater application of

**Figure 1.** (*a*) Illustration of the VFD which houses a rapidly rotating borosilicate glass tube (20 mm OD, 17.5 mm ID), operating under an inert atmosphere of nitrogen gas. Photograph of the solution obtained (*b*) under confined mode of operation of the VFD (30 min) and (*c*) continuous flow, flow rate 0.5 ml min$^{-1}$. Both modes were optimized at 4000 r.p.m., $\theta$ 45°, for a concentration of MXene 0.5 mg ml$^{-1}$ in a 1 : 1 ratio of IPA and water.

the MXene relative to MAX phases [12]. The work function of MXene sheets and those functionalized with F, OH and O groups have been studied using first-principles calculations [9]. The bandgap of MXene sheets are ultra-low when the sheets are –OH functionalized [9,13]. There is also a work function dependence on the transition metal and associated charge transfer between the functional groups and the substrate, and overall changes in the total surface dipole moment [10].

The exfoliation (delamination) of MXenes provides unique material with different functional groups, microstructure and morphology, which impacts on electrochemical response of the material. The exfoliation of MXene requires harsh chemicals or surfactant in a number of different methods, as in sonication [7,14], heating [15,16] or both sonication and heating [17], and the use of electric fields [18,19]. Interestingly, sonication can result in different shaped material [14,20] presumably arising from extreme localized conditions associated with cavitation. In the present study, we explore the utility of the vortex fluidic device (VFD), figure 1, for exfoliating Ti$_2$CT$_x$ type MXene in the absence of harsh chemicals or surfactant. The VFD microfluidic platform delivers mechanoenergy in the dynamic film. We hypothesized that it would be effective in exfoliating the 2D material, based on its success in exfoliating graphene and *h*-BN [21–23]. The resulting exfoliation is at room temperature as a single-step process in a mixture of isopropyl alcohol (IPA) and water (ratio 1 : 1), under continuous flow conditions (see below), and under an inert atmosphere of nitrogen gas to avoid oxidation of the MXene [17].

The vortex fluidic device (VFD) [21,24–26], figure 1, has a borosilicate glass tube (OD 20 mm, ID 17.5 mm and 19.5 cm in length) open at one end. The operating parameters of the device are then systematically explored for optimizing any process. The tube is rotated at high speed (up to 9000 r.p.m.) and can be inclined at an angle, $\theta$, of −45° to 90° relative to the horizontal position, with +45° being optimal for a large number of applications, including in the present study. The VFD has two types of processing—confined mode and continuous flow. The confined mode is where a finite volume of liquid is placed in the glass tube which is tilted at 45° and spun at high speed for a designated period, figure 1*b*. This mode of operation of the VFD has proved invaluable in optimizing any processing in translating it into continuous flow where liquid is delivered to the inside of the rotating tube at a controlled flow rate, figure 1*c* [27]. The utility of processing in the VFD, beyond the aforementioned exfoliation of graphene from graphite and *h*-BN [21–23], covers scrolling graphene [23] and graphene oxide [28], protein folding [29], enhancing enzymatic reactions [24], controlling organic synthesis [25,30], probing the structure of self-organized systems [31], protein separation [32] and more [33–37].

## 2. Experimental procedure

### 2.1. Materials

Ti$_2$AlC powder precursor (MAX) was obtained commercially (Kanthal, Maxthal 211 Ti$_2$AlC). The etching of the aluminium from Ti$_2$AlC was conducted in a 20% aqueous hydrofluoric acid (HF) solution (Sigma

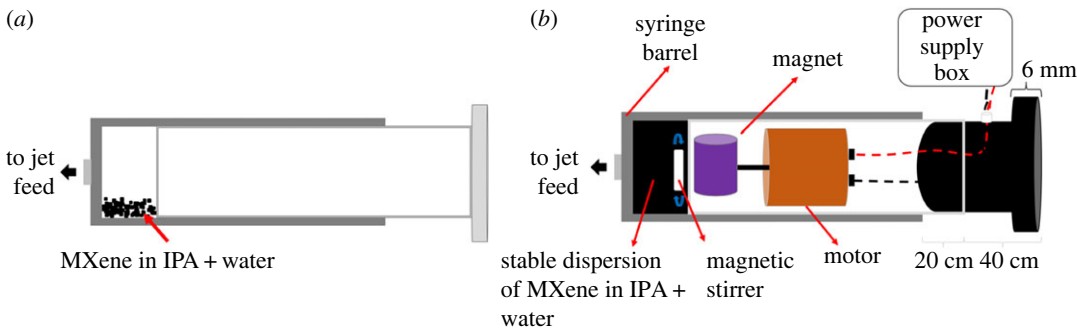

**Figure 2.** (a) Normal syringe housing an unstable dispersion of MXene in IPA and water. (b) Syringe housing a magnetic stirrer driven by an electric motor inside the plunger.

Aldrich) for 24 h at room temperature. The resulting suspension was filtered and washed with deionized water (DI) to reach a pH > 6 [14]. This as-prepared MXene was dispersed in distilled IPA purchased from Sigma-Aldrich and Milli-Q water.

## 2.2. Synthesis of MXene nanoparticles and exfoliation of MXene

MXene was dispersed in distilled IPA and Milli-Q water using sonication (25 min, 6 kHz). The mixture was then transferred to an in-house developed magnetically stirred syringe, figure 2 [35,38]. The rapidly stirred dispersion of MXene in IPA and water (0.5 mg ml$^{-1}$) was then delivered using a jet feed to the base of the rapidly rotating 20 mm OD glass tube in the VFD with another jet feed delivery a low flow rate of dry nitrogen gas. The tube was inclined at 45° and spun at a pre-determined speed, which was optimized at 4000 r.p.m., with the optimized flow rate at 0.5 ml min$^{-1}$, resulting in ca 7% exfoliated MXene and fragmentation to nanoparticles, exiting the tube. We refer to the exfoliated MXene and MXene NPs exiting the tube during continuous flow processing as 'collected'. After 1 h of processing (30 ml passed through the VFD), 3.5 ml remained in the tube which was added to 3 ml of a 1 : 1 mixture of IPA and water. The mixture was centrifuged (700g) for 3 min to separate large MXene particles. The supernatant afforded ca 8% exfoliated MXene sheets and MXene nanoparticles. This material which builds up in the VFD during the continuous flow processing is referred to as 'retained' material.

## 2.3. Characterization

MXene sheets and MXene nanoparticles were characterized using scanning electron microscopy (SEM) (Inspect FEI F50 SEM), atomic force microscopy (AFM—Nanoscope 8.10 tapping mode), Raman spectroscopy (WiTec Alpha 300R $\lambda_{exc}$ = 532 nm), XRD (Bruker D8 Advance Eco, Co–K$\alpha$, $\lambda$ = 1.7889 Å), ATR-FTIR Perkin Elmer Frontier, TEM (FEI Tecnai F20 operated at 120 kV), UV–Vis spectrophotometer Varian Cary 50 EST 70772 and dynamic light scattering (DLS) (Zetasizer Nanoseries nano-zs, Malvern).

# 3. Results and discussion

The VFD is a flexible processing platform, with a number of operating parameters (rotational speed, tilt angle, $\theta$, (figure 1) and flow rates) to be systematically explored, along with the choice of solvent. The use of a number of different solvents were investigated, namely water, NMP (N-methyl-2-pyrrolidone), IPA, DMF (dimethylformamide), o-xylene, m-xylene and toluene. However, their use resulted in little or no exfoliated MXene or resulted in the formation of nanoparticles. The same was also found for combinations of solvents, in a 1 : 1 ratio, including NMP with water, DMF with o-xylene, DMF with toluene, DMF with IPA, toluene with IPA, m-xylene with IPA, o-xylene with IPA and IPA with water, for different speeds and different concentrations, electronic supplementary material, figures S1 and S2 [39]. Considering the use of co-solvents relates to their success in a number of process, including forming graphene scrolls directly from graphite [23]. In addition, irradiating the material *in situ* in the VFD was explored using a Nd:YAG pulsed laser operating at 1064 nm at different power settings, as another parameter to enhance the exfoliation, using a mixture of IPA and water with a volume ratio of 1 : 1, electronic supplementary material, figure S3. Varying the tilt angle $\theta$ was also explored, additional to the common tilt angle of +45°, including 20°, −20° and −45°, while varying the rotational speed of the tube,

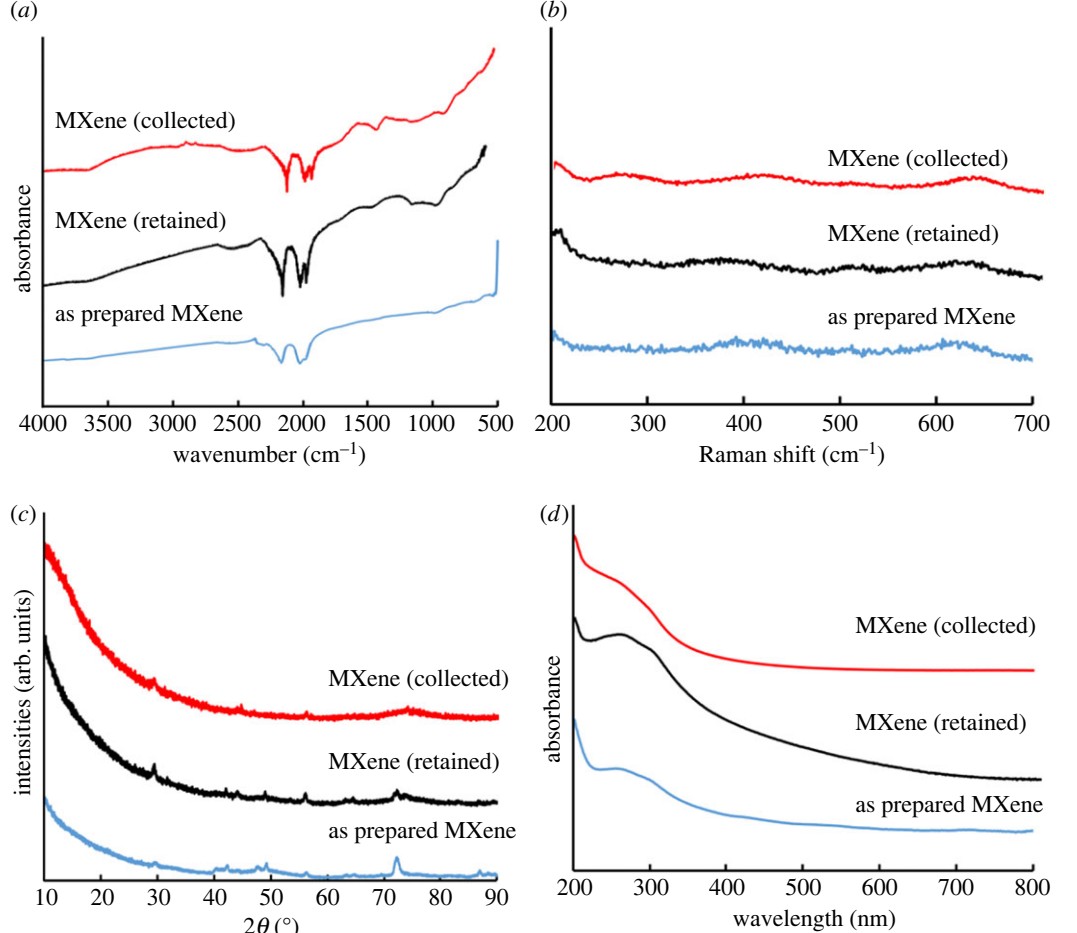

**Figure 3.** Exfoliated MXene prepared using VFD processing under $N_2$ gas, $\theta$ 45°, concentration 0.5 mg ml$^{-1}$ in IPA and water (1 : 1), rotational speed 4000 r.p.m. and flow rate 0.5 ml min$^{-1}$. (*a*) ATR-FTIR spectra, (*b*) Raman spectra, (*c*) PXRD and (*d*) UV–Vis spectroscopy.

electronic supplementary material, figure S4. We note that −45° tilt angle was effective in exfoliating *h*-BN in water [22]. Varying the different operating parameters of the VFD led to determining the optimal conditions for the exfoliation of MXene and the highest yield of material exiting (collected) the tube under continuous flow. This was for a tilt angle of 45°, which is consistent with most optimized processes in the VFD.

A colour change was observed for the suspension of MXene (from dark grey to yellow) after processing in the VFD in air, which is not surprising, given that the material is sensitive to air, being readily oxidized [14,17]. Thus, all experiments were subsequently done under an inert atmosphere of nitrogen to circumvent any oxidation arising from high uptake of molecular oxygen in the thin film in the VFD. Electronic supplementary material, figures S5 and S6 summarize experiments done under $N_2$ gas using different solvents. These include water, DMF, ethanol and IPA, and mixtures of solvents, including IPA and water, DMF and water, ethanol and water and DMF and *o*-xylene. IPA and water, volume ratio 1 : 1 was deemed to be the optimal solvent, and was used for subsequent experiments. The optimized processing is with respect to the highest level of exfoliation and the yield of material exiting the tube, electronic supplementary material, figure S14(a) [39].

We found that the confined mode of operation of the VFD gave little exfoliation of the MXene, unlike continuous flow, and its use was adhered to for all subsequent experiments, electronic supplementary material, figure S7. This included varying the rotational speed, flow rate and concentration of MXene in the 1 : 1 mixture of IPA and water. We found 4000 r.p.m. gave the highest level of exfoliated MXene, with some fragmentation into small nanoparticles of the material; the combined exfoliated MXene and nanoparticles accounted for *ca* 7% of the starting material. The two materials were recalcitrant towards separation by centrifugation. Rotational speeds of 5000, 6000, 7000 and 8000 r.p.m. gave little exfoliation at the ubiquitous optimal 45° tilt angle, electronic supplementary material, figures S8, S9 and S14(b). The optimal flow rate of MXene in IPA and water was 0.5 ml min$^{-1}$, with a lower flow rate,

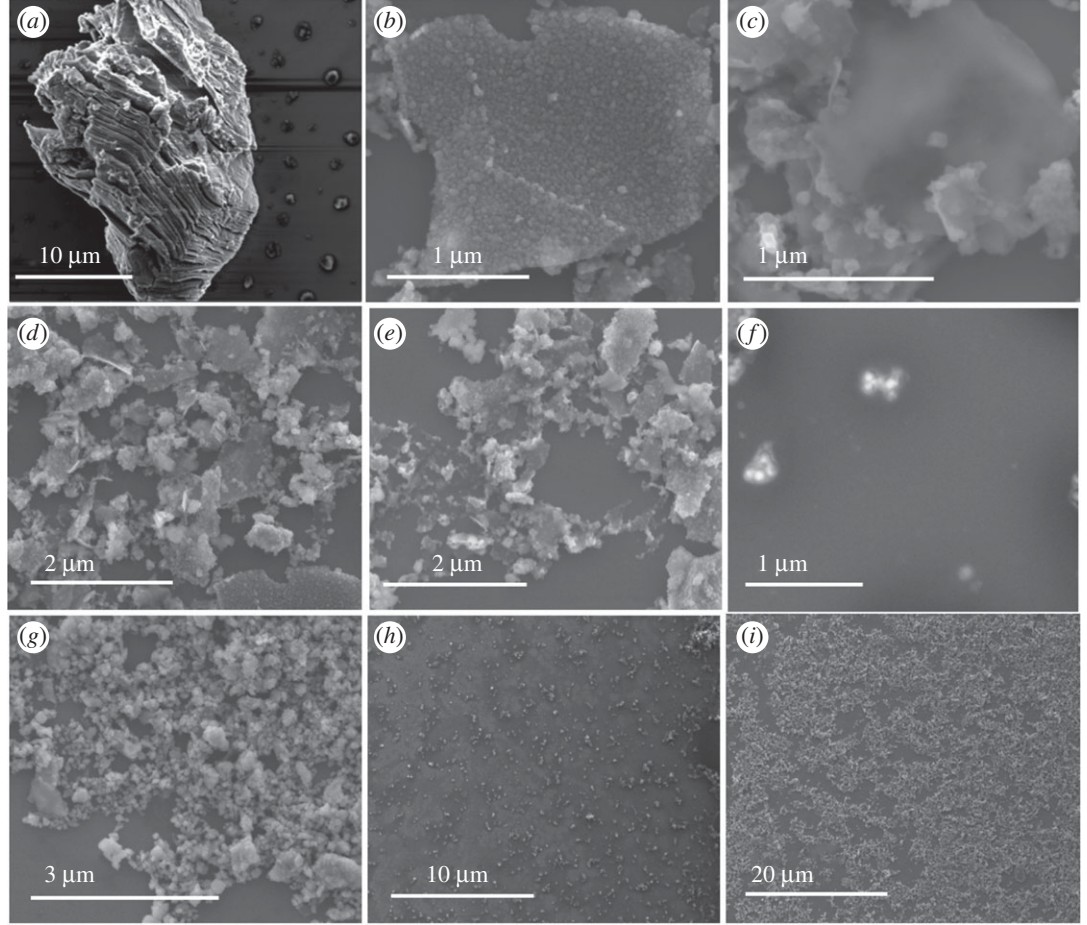

**Figure 4.** SEM images for MXene (collected) drop cast on a silicon wafer, post-VFD processing under $N_2$ gas, $\theta$ 45°, concentration 0.5 mg ml$^{-1}$ in IPA and water (1:1), rotational speed 4000 r.p.m. and flow rate 0.5 ml min$^{-1}$. (a) MXene as prepared, (b–e) exfoliated MXene, (f) backscattering of MXene nanoparticles and (g–i) MXene nanoparticles.

0.25 ml min$^{-1}$, and a higher flow rate, 0.75 ml min$^{-1}$, not as effective. For all these flow rates, the concentrations of MXene in IPA and water (1:1) were varied, at 0.1, 0.25, 0.5 and 1.0 mg ml$^{-1}$. The optimal concentration was found to be 0.5 mg ml$^{-1}$, with other concentrations resulting in lower yields, electronic supplementary material, figure S14(c). Thus, the overall optimum processing parameters for generating exfoliated MXene in a single pass in the VFD are a concentration of the material in a 1:1 ratio of IPA and water at 0.5 mg ml$^{-1}$, flow rate 0.5 ml min$^{-1}$ and 4000 r.p.m. rotational speed. In addition, after VFD processing under continuous flow for 1 h, an additional *ca* 8% of both exfoliated MXene and MXene nanoparticles were collected from the tube (retained), electronic supplementary material, figure S15. We initially used SEM images of drop cast material on silicon wafers to ascertain the effect of varying the rotation speed, flow rate and tilt angle of the VFD, along with the choice of solvent, for the highest degree of the exfoliation of MXene. We established the structure of MXene (retained) using ATR-FTIR spectra, Raman spectroscopy, powder X-ray diffraction (PXRD) and UV–Vis spectroscopy, figure 3.

The shear stress in the VFD is effective in both exfoliating and fragmenting MXene. This is consistent with other established top-down nanomaterials syntheses in the device, covering exfoliating [21,22] and scrolling 2D materials [22,23,28], slicing carbon nanotubes [26,40] and more [33–35,41].

Figure 4f shows backscattering SEM to determine the elements present in the product relative to the starting material, for different locations, figure 4g–i (collected) and electronic supplementary material, figure S10(a–d) (retained). TEM images established the presence of some MXene nanoparticles, figure 5a–d and electronic supplementary material, figure S11, with figure 6a,b providing additional TEM, and SEM images, respectively, of nanospheres. AFM images provided additional information on the shape of the nanospheres, in measuring their height. The results are in agreement with those from TEM and SEM (figure 6c,e). DLS of solutions of MXene nanoparticles in IPA and water (1:1) generated in the VFD after processing gave the average diameter of nanospheres as 68 nm, figure 6d. SEM images

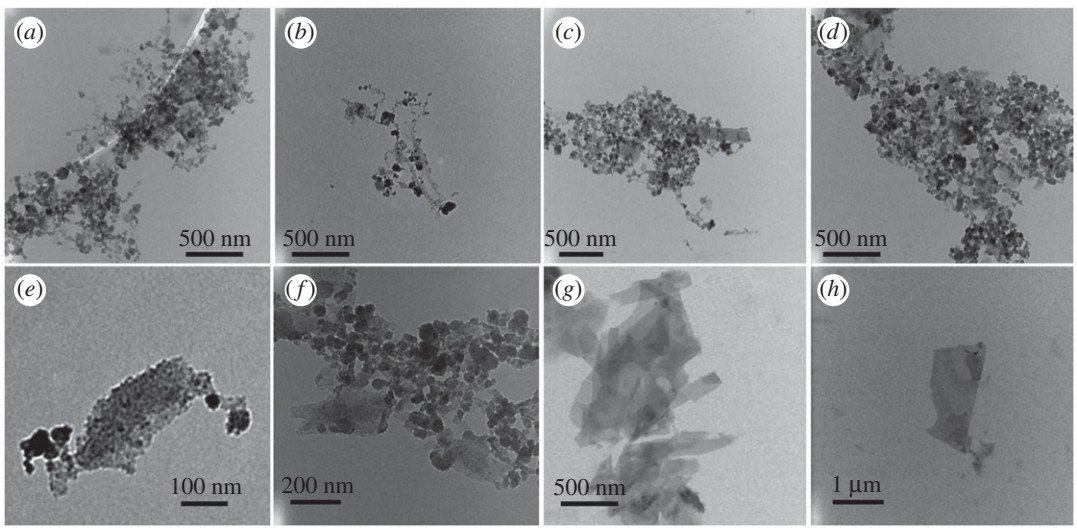

**Figure 5.** TEM images for MXene drop cast on grid, post-VFD processing under $N_2$ gas, $\theta$ 45°, concentration 0.5 mg ml$^{-1}$, in IPA and water (1 : 1), rotational speed 4000 r.p.m. and flow rate 0.5 ml min$^{-1}$.

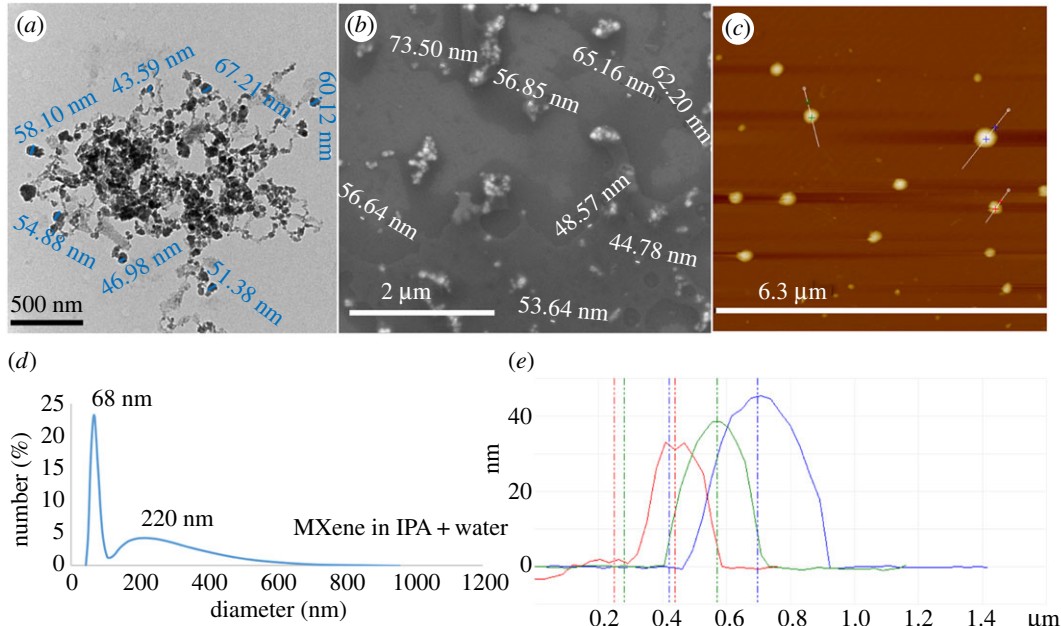

**Figure 6.** MXene nanoparticles formed during VFD processing under $N_2$ gas, with $\theta$ 45°, concentration 0.5 mg ml$^{-1}$ in IPA and water (1 : 1), rotational speed 4000 r.p.m. and flow rate 0.5 ml min$^{-1}$. (a) TEM image of MXene nanoparticles drop cast on a grid. (b) SEM image of MXene nanoparticles drop cast on a silicon wafer. (c) AFM image of MXene nanoparticles drop cast on a silicon wafer. (d) DLS of exfoliated MXene generated in IPA and water (1 : 1). (e) Height of MXene nanoparticles as determined using AFM images.

established the presence of exfoliated MXene, figure 4b–e, electronic supplementary material, figure S10(e–l), with AFM images establishing the mean thickness of VFD-exfoliated MXene sheets as 3 nm, figures 7 and 8c, electronic supplementary material, figure S12.

The TEM images were used to determine the size and number of MXene sheets generated using the VFD processing, figure 5e–h, along with the size of the spheres, electronic supplementary material, figure S8(a–e). However, the spheres were found to be unstable, with the drop cast spheres collapsing after 3 days. Interestingly, the spheres have holes in them, electronic supplementary material, figure S8(f,g), but the origin of these is unclear.

ATR-FTIR for MXene has characteristic peaks, for material exiting the tube during processing in the VFD, and material retained in the tube, and as-prepared material, with peaks at 684 and 991 cm$^{-1}$ for Ti–O and C–F, respectively [42]. However, the VFD-processed MXene has a peak at 3674 and 1499 cm$^{-1}$, corresponding to

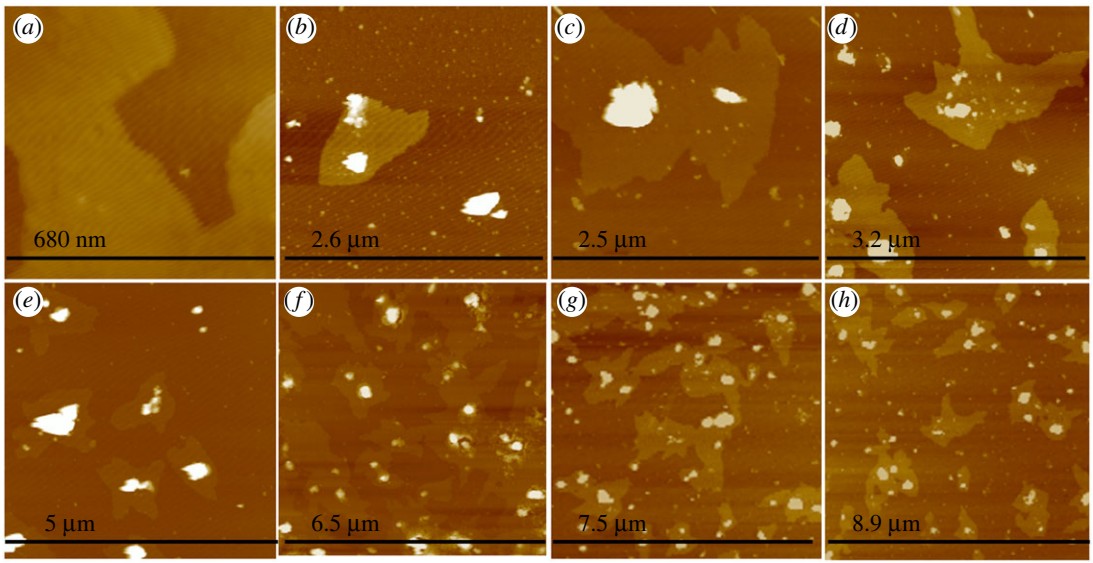

**Figure 7.** AFM images for MXene drop cast on silicon wafers, post-VFD processing under $N_2$ gas, $\theta$ 45°, concentration 0.5 mg ml$^{-1}$ in IPA and water (1 : 1), rotational speed 4000 r.p.m. and flow rate 0.5 ml min$^{-1}$.

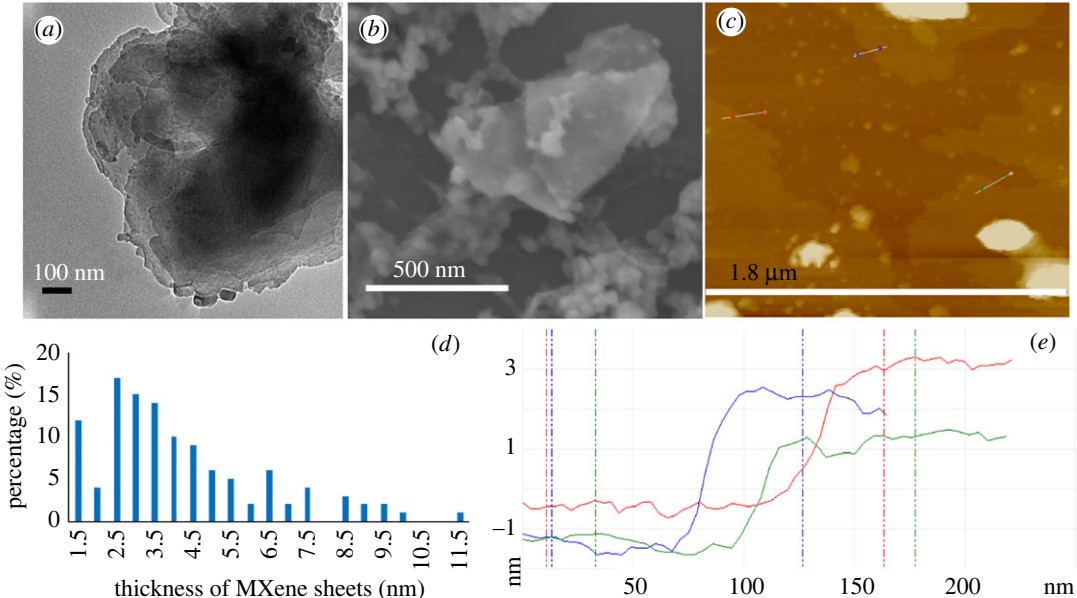

**Figure 8.** Exfoliated MXene generated during VFD processing under $N_2$ gas, $\theta$ 45°, concentration 0.5 mg ml$^{-1}$ in IPA and water (1 : 1), rotational speed 4000 r.p.m. and flow rate 0.5 ml min$^{-1}$. (*a*) TEM image for MXene sheets drop cast on a grid. (*b*) SEM image of MXene sheets drop cast on a silicon wafer. (*c*) AFM image of MXene sheets drop cast on a silicon wafer. (*d*) Count of MXene sheets from AFM images. (*e*) Height of MXene sheets from AFM images.

O–H, respectively, of solvent, figure 3*a* [43–45]. Raman spectra of MXene as prepared have three peaks at 259, 421 and 607 cm$^{-1}$. The material exiting the VFD tube has peaks at 263, 419 and 613 cm$^{-1}$, and retained in the tube at 250, 405 and 613 cm$^{-1}$. The slight shift of Raman peaks for exiting and retained material presumably relates to the presence of nanospheres, along with the exfoliated MXene, only for material exiting the tube, figure 3*b* [14,46]. Powder XRD of MXene confirms the presence of Ti$_2$CT$_x$ MXenes, which has hexagonal P6$_3$/mmc symmetry with a lattice constant $a \approx 0.3$ nm [47,48]. The $2\theta$ peaks (Co–K$\alpha$, $\lambda = 1.7889$ Å) of MXene as prepared were 29.4°, 40.6°, 42.4° and 72.4°, with corresponding values for material exiting the tube 29.4°, 40.6°, 42.4° and 72.4°, figure 3*c* [49,50]. The small shift after VFD processing is associated with a reduction in the thickness of the MXene sheets along with fragmentation. Finally, the UV–Vis spectra of MXene as prepared and the materials produced herein are similar, figure 3*d* [51].

# 4. Conclusion

We have developed a continuous process for exfoliating 2D-MXene into multi-layered sheets *ca* 3 nm thick, driven by the mechanoenergy in the dynamic thin film in the VFD. During this processing, we also established that the VFD can fragment the laminar MXene into nanoparticles (average diameter 68 nm). The optimized parameters for processing the MXene under an inert atmosphere of nitrogen gas were for a concentration of MXene at 0.5 mg ml$^{-1}$ in a 1:1 mixture of IPA and water, with a rotational speed of 4000 r.p.m. and flow rate of 0.5 ml min$^{-1}$. The exfoliated material and nanoparticles of the material have potential in applications such as environmental remediation [45], biological application [2–5], electronic devices [9,10], supercapacitors [10,11] and next generation of batteries. [6,7] Understanding the complex fluid dynamics in the thin film in the VFD is a major challenge currently being tackled. The choice of a single solvent or mixed solvent system depends on the application, as does the effect of different operating parameters of the device (tilt angle, rotation speed and flow rate of solvent), which need to be systematically explored.

Data accessibility. This article has no additional data.

Authors' contributions. A.H.M.A. carried out characterization and synthesis, S.K. prepared the MXene and A.H.M.A., S.K., N.P.P., W.D.L. and C.L.R. contributed to the writing the manuscript.

Competing interests. We declare we have no competing interests.

Funding. The authors gratefully acknowledge financial support from the Iraq Government, Ministry of Higher Education and Scientific Research, and the Australian Research Council and the Government of South Australia. S.K. and N.P.P. (BOE-SAS grant) gratefully acknowledge the financial support provided by LSBU.

Acknowledgements. The use of facilities in the Australian Microscopy & Microanalysis Research Facility (AMMRF) and the Australian National Fabrication Facility (ANFF) at the South Australian nodes of the AMMRF and ANFF under the National Collaborative Research Infrastructure Strategy are acknowledged.

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
