## [Reviewer comments · Royal Society Open Science]

Review History

RSOS-192255.R0 (Original submission)

Review form: Reviewer 1

Is the manuscript scientifically sound in its present form?

No

Are the interpretations and conclusions justified by the results?

Yes

Is the language acceptable?

Yes

Do you have any ethical concerns with this paper?

No

Have you any concerns about statistical analyses in this paper?

No

Recommendation?

Major revision is needed (please make suggestions in comments)

Comments to the Author(s)

Mxene nanosheets exfoliated from Mxene precursor hold great application potential in areas such as biology, batteries, electronics, supercapacitors, etc. However, their efficient exfoliation remains a challenge. This work provides an unique strategy for exfoliating them from Mxene precursor by utilizing a vortex fluidic device with optimised conditions. I recommendate its publication if the following revisions are made on the present basis.

- 1) The whole experimental course is not clear, please design a complex schematic diagram to elucidate the exfoliation and separation course of Mxene nanosheets.
- 2) The effects of various process parameters on the exfoliation efficiencies of mxene nanosheets should be quantitatively discussed.
- 3) Please provide results on lateral sizes and distribution of resulting mxene sheets.
- 4) Please give explanation on the significant influence from the tilte angle?

Review form: Reviewer 2 (Kabeer Jasuja)

Is the manuscript scientifically sound in its present form?

No

Are the interpretations and conclusions justified by the results?

No

Is the language acceptable?

Yes

Do you have any ethical concerns with this paper?

No

Have you any concerns about statistical analyses in this paper?

No

Recommendation?

Major revision is needed (please make suggestions in comments)

Comments to the Author(s)

The manuscript is presenting a useful and scalable approach to exfoliated a MAX phase compound. However, there are several places where the manuscript can be improved. These are listed below:

1. It would be helpful if the authors can further explain the statement that sonication can result in different shaped material.
2. The insets shown in the Fig 1b and Fig 1c can be improved for their clarity, and their associated message.
3. The manuscript would benefit if some statements are made shorter and rewritten as 2-3 shorter statements. For example, the statements at the end of the introduction section are very long. The authors are requested to revisit the manuscript and see if this correction can be made at other places as well within the manuscript.
4. The authors have tested co-solvent approach in addition to using single solvents. It would be helpful if the authors can include the thought process of testing co-solvents and by citing the relevant literature.

5. Figure 3 compares spectra of three types of MXene samples - collected, retained, and as prepared. This figure can be improved at several fronts:

(a) From the corresponding text on Page 3, It is not being clear what the phrases collected, retained, and as prepared refer to. Should the readers expect a difference in these spectra.

(b) The features of these spectra can be explained at the right places in the manuscript. At present, the spectra are being explained after the discussion for Figure 4. For example, the Rama spectra can be explained further, the absorbance peaks can also be explained, and if there are some differences, these can be commented upon. The manuscript can be reorganized to make it in sync with the order in which figures are appearing.

(c) The y-axis of XRD spectra doesn't have a corresponding axis title. The spectra for collected/retained is different from as prepared. What could be some reasons behind this?

6. The FESEM images g-i shown in Figure 4 are not clear. The nanoparticulate formation is observed more clearly in the TEM images, and hence the authors can use this for supporting their claims. The authors should also acknowledge the formation of nanoparticulate matter in their title as these are not necessarily 2D.

7. The labels shown in Figure 6a and 6b can be made more clear, these are not legible in the present form. Similarly Fig 6e and Fig 8e can be improved as well.

8. The title of the paper states exfoliation of 2D Mxene. The title should be revised as exfoliation happens of a layered material, which is Ti_2AlC in this case. Also, in addition to nanosheets, there are also nanoparticulate matter being formed. This should also be incorporated in the title.

Decision letter (RSOS-192255.R0)

24-Feb-2020

Dear Professor Raston:

Title: Continuous flow vortex fluidic mediated exfoliation of 2D MXene
Manuscript ID: RSOS-192255

The editor assigned to your manuscript has now received comments from reviewers. We would like you to revise your paper in accordance with the referee and Subject Editor suggestions which can be found below (not including confidential reports to the Editor). Please note this decision does not guarantee eventual acceptance.

Please submit your revised paper before 18-Mar-2020. Please note that the revision deadline will expire at 00.00am on this date. If we do not hear from you within this time then it will be assumed that the paper has been withdrawn. In exceptional circumstances, extensions may be possible if agreed with the Editorial Office in advance. We do not allow multiple rounds of revision so we urge you to make every effort to fully address all of the comments at this stage. If deemed necessary by the Editors, your manuscript will be sent back to one or more of the original reviewers for assessment. If the original reviewers are not available we may invite new reviewers.

RSC Associate Editor:
Comments to the Author:
(There are no comments.)

RSC Subject Editor:
Comments to the Author:
(There are no comments.)

Reviewers' Comments to Author:
Reviewer: 1

Comments to the Author(s)

Mxene nanosheets exfoliated from Mxene precursor hold great application potential in areas such as biology, batteries, electronics, supercapacitors, etc. However, their efficient exfoliation remains a challenge. This work provides a unique strategy for exfoliating them from Mxene precursor by utilizing a vortex fluidic device with optimised conditions. I recommendate its publication if the following revisions are made on the present basis.

- 1) The whole experimental course is not clear, please design a complex schematic diagram to elucidate the exfoliation and separation course of Mxene nanosheets.
- 2) The effects of various process parameters on the exfoliation efficiencies of mxene nanosheets should be quantitatively discussed.
- 3) Please provide results on lateral sizes and distribution of resulting mxene sheets.

4) Please give explanation on the significant influence from the tilte angle?

Reviewer: 2

Comments to the Author(s)

The manuscript is presenting a useful and scalable approac to exfoliated a MAX phase compound. However, there are several places where the manuscript can be improved. These are listed below:

1. It would be helpful if the authors can further explain the statement that sonication can result in different shaped material.
2. The insets shown in the Fig 1b and Fig 1c can be improved for their clarity, and their associated message.
3. The manuscript would benefit if some statements are made shorter and rewritten as 2-3 shorter statements. For example, the statements at the end of the introduction section are very long. The authors are requested to revisit the manuscript and see if this correction can be made at other places as well within the manuscript.
4. The authors have tested co-solvent approach in addition to using single solvents. It would be helpful if the authors can include the thought process of testing co-solvents and by citing the relevant literature.
5. Figure 3 compares spectra of three types of MXene samples - collected, retained, and as prepared. This figure can be improved at several fronts:
 - (a) From the corresponding text on Page 3, It is not being clear what the phrases collected, retained, and as prepared refer to. Should the readers expect a difference in these spectra.
 - (b) The features of these spectra can be explained at the right places in the manuscript. At present, the spectra are being explained after the discussion for Figure 4. For example, the Rama spectra can be explained further, the absorbance peaks can also be explained, and if there are some differences, these can be commented upon. The manuscript can be reorganized to make it in sync with the order in which figures are appearing.
 - (c) The y-axis of XRD spectra doesn't have a corresponding axis title. The spectra for collected/retained is different from as prepared. What could be some reasons behind this?
6. The FESEM images g-i shown in Figure 4 are not clear. The nanoparticulate formation is observed more clearly in the TEM images, and hence the authors can use this for supporting their claims. The authors should also acknowledge the formation of nanoparticulate matter in their title as these are not necessarily 2D.
7. The labels shown in Figure 6a and 6b can be made more clear, these are not legible in the present form. Similarly Fig 6e and Fig 8e can be improved as well.
8. The title of the paper states exfoliation of 2D Mxene. The title should be revised as exfoliation happens of a layered material, which is Ti₂AlC in this case. Also, in addition to nanosheets, there are also nanoparticulate matter being formed. This should also be incorporated in the title.

Author's Response to Decision Letter for (RSOS-192255.R0)

See Appendix A.

RSOS-192255.R1 (Revision)

Review form: Reviewer 1

Is the manuscript scientifically sound in its present form?

Yes

Are the interpretations and conclusions justified by the results?

Yes

Is the language acceptable?

Yes

Do you have any ethical concerns with this paper?

No

Have you any concerns about statistical analyses in this paper?

No

Recommendation?

Accept as is

Comments to the Author(s)

The revision has been made carefully according to the comments from reviewers. So I am glad to recommend its publication in the present state.

Decision letter (RSOS-192255.R1)

30-Mar-2020

Dear Professor Raston:

Title: Continuous flow vortex fluidic mediated exfoliation of 2D MXene
Manuscript ID: RSOS-192255.R1

It is a pleasure to accept your manuscript in its current form for publication in Royal Society Open Science. The chemistry content of Royal Society Open Science is published in collaboration with the Royal Society of Chemistry.

RSC Associate Editor:
Comments to the Author:
(There are no comments.)

RSC Subject Editor:
Comments to the Author:
(There are no comments.)

Reviewer(s)' Comments to Author:
Reviewer: 1

Comments to the Author(s)
The revision has been made carefully according to the comments from reviewers. So I am glad to recommendate its publication in the present state.

Appendix A

Cover letter

Dear Editor,

Thank you for your positive feedback. The manuscript has been revised accordingly. Please refer to our responses below, in addressing all issues raised by the referee, as well as your comments. We look forward to hearing from you.

Thank you,

Professor Colin Raston

Reviewer: 1

1- The whole experimental course is not clear, please design a complex schematic diagram to elucidate the exfoliation and separation course of MXene nanosheets.

Answer: We have been added two schematic diagrams to the revised manuscript and revised SI (Figure S14 and S15) to explain the processing of MXene, for MXene nanosheets and NPs MXene exiting the VFD, and MXene nanosheets and NPs of MXene retained in the VFD tube.

2- The effects of various process parameters on the exfoliation efficiencies of MXene nanosheets should be quantitatively discussed.

Answer: A paragraph has been added to the revised manuscript (page 4) to clarify the quantitatively discussed “**We initially used SEM images of drop cast material on silicon wafers to ascertain the effect of varying the rotation speed, flow rate, and tilt angle of the VFD, along with the choice of solvent, for the highest degree of exfoliation of MXene.**”

3- Please provide results on lateral sizes and distribution of resulting MXene sheets.

Answer: We have provided the lateral size of exfoliated material, as derived from SEM images, in Figure S1-S7.

4- Please give explanation on the significant influence from the tilt angle?

Answer: A following has been added to the revised manuscript (page 3) to clarify the tilt angle “**Varying the different operating parameters of the VFD led to determining the optimal conditions for exfoliation of MXene and the highest yield of material exiting (collected) the tube under continuous flow. This was for a tilt angle of 45°, which is consistent with most optimised processes in the VFD.**”

Reviewer: 2

1- It would be helpful if the authors can further explain the statement that sonication can result in different shaped material.

Answer: This has been addressed, with inclusion of the following statement, page (1): “presumably arising from extreme localised conditions associated with cavitation”.

2- The insets shown in the Fig 1b and Fig 1c can be improved for their clarity, and their associated message.

Answer: This has been addressed on page (2), by the following “The VFD has two types of processing – confined mode and continuous flow. The confined mode is where a finite volume of liquid is placed in the glass tube which is tilted at 45° and spun at high speed for a designated period, Fig. 1(b). This mode of operation of the VFD has proved invaluable in optimising any processing in translating it into continuous flow where liquid is delivered to the inside of the rotating tube at a controlled flow rate, Fig. 1(c).²⁷”

3. The manuscript would benefit if some statements are made shorter and rewritten as 2-3 shorter statements. For example, the statements at the end of the introduction section are very long. The authors are requested to revisit the manuscript and see if this correction can be made at other places as well within the manuscript.

Answer: Changes have been made at the end of the introduction section in the revised manuscript, and a number of other parts of the manuscript, which have been heightened.

4. The authors have tested co-solvent approach in addition to using single solvents. It would be helpful if the authors can include the thought process of testing co-solvents and by citing the relevant literature.

Answer: We have added the following statement in addressing this on page (3), along with the inclusion of ref. 39 “Considering the use of co-solvents relates to their success in a number of process, including forming graphene scrolls directly from graphite.²³”

5. Figure 3 compares spectra of three types of MXene samples - collected, retained, and as prepared. This figure can be improved at several fronts:

(a) From the corresponding text on Page 3, It is not being clear what the phrases collected, retained, and as prepared refer to. Should the readers expect a difference in these spectra.

Answer: Sentences have been added to the revised manuscript (page 2) to clarify the meaning of collected, retained, and as prepared **“We refer to the exfoliated MXene and MXene NPs exiting the tube during continuous flow processing as ‘collected’”** and **“This material which builds up in the VFD during the continuous flow processing is referred to as ‘retained’ material..”**

Also, we have inserted a sentence to explain the difference between the spectra (page 4) **“The small shift after VFD processing is associated with a reduction in the thickness of the MXene sheets along with fragmentation.”**

(b) The features of these spectra can be explained at the right places in the manuscript. At present, the spectra are being explained after the discussion for Figure 4. For example, the Rama spectra can be explained further, the absorbance peaks can also be explained, and if there are some differences, these can be commented upon. The manuscript can be reorganized to make it in sync with the order in which figures are appearing.

Answer: We explained the images for SEM, TEM and AFM as a unified approach, then we talked about the spectra separately to integrate the multiple techniques used to characterise the material.

(c) The y-axis of XRD spectra doesn't have a corresponding axis title. The spectra for collected/retained is different from as prepared. What could be some reasons behind this?

Answer: We have added intensities (a.u.) to y-axis of the XRD pattern and have explained the reason for in response to (5(a), above, with the inclusion of an additional text (page 4).

6. The FESEM images g-i shown in Figure 4 are not clear. The nanoparticulate formation is observed more clearly in the TEM images, and hence the authors can use this for supporting their claims. The authors should also acknowledge the formation of nanoparticulate matter in their title as these are not necessarily 2D.

Answer: We have been changed the SEM images in Figure 4(g-i) to be more relative with TEM images, and the title of the manuscript has been changed to highlight fragmentation as well.

7. The labels shown in Figure 6a and 6b can be made more clear, these are not legible in the present form. Similarly Fig 6e and Fig 8e can be improved as well.

Answer: Changes have been made in Figure 6a,6b,6e and Figure 8e to make them clear.

8. The title of the paper states exfoliation of 2D Mxene. The title should be revised as exfoliation happens of a layered material, which is Ti₂AlC in this case. Also, in addition to nanosheets, there are also nanoparticulate matter being formed. This should also be incorporated in the title.

Answer: the title has been changed to be “Continuous flow vortex fluidic mediated exfoliation and fragmentation of 2D MXene.”

Reference//

- [27] J. Britton, K. A. Stubbs, G. A. Weiss, C. L. Raston, *Chemistry–A European Journal* **2017**, *23*, 13270-13278.
- [39] S. Seyedin, J. Zhang, K. A. S. Usman, S. Qin, A. M. Glushenkov, E. R. S. Yanza, R. T. Jones, J. M. Razal, *Adv. Sci. News*, 2019, *3*, 1900037.